# In Situ Preparation of Amphibious ZnO Quantum Dots with Blue Fluorescence Based on Hyperbranched Polymers and their Application in Bio-Imaging

**DOI:** 10.3390/polym12010144

**Published:** 2020-01-06

**Authors:** Gaiying Lei, Shu Yang, Ranran Cao, Peng Zhou, Han Peng, Rui Peng, Xiaoming Zhang, Yujiao Yang, Yueyang Li, Mengyue Wang, Yaru He, Linzhu Zhou, Jimin Du, Weimin Du, Yunfeng Shi, Hankui Wu

**Affiliations:** 1School of Chemistry and Chemical Engineering, Henan Normal University, Xinxiang 453007, China; lgying1006@163.com; 2School of Chemistry and Chemical Engineering, Anyang Normal University, Anyang 455000, China; Y15736967669@163.com (S.Y.); c15864418126@163.com (R.C.); z181001066@163.com (P.Z.); ph18337278759@163.com (H.P.); pr15896849973@163.com (R.P.); 18337278676@163.com (X.Z.); y15042611565@163.com (Y.Y.); 18637818665@163.com (Y.L.); myw0618@163.com (M.W.); heyaru1999@163.com (Y.H.); djm@aynu.edu.cn (J.D.); dwmchem@163.com (W.D.); 3Henan Province Key Laboratory of New Optoelectronic Functional Materials, Anyang Normal University, Anyang 455000, China; 4School of Chemistry and Chemical Engineering, Shanghai Jiao Tong University, 800 Dongchuan Road, Shanghai 200240, China; linzhu.zhou@hotmail.com

**Keywords:** hyperbranched polymers, amphibious ZnO quantum dots, blue fluorescence, bio-imaging

## Abstract

A new strategy for preparing amphibious ZnO quantum dots (QDs) with blue fluorescence within hyper-branched poly(ethylenimine)s (HPEI) was proposed in this paper. By changing [Zn^2+^]/[OH^−^] molar ratio and heating time, ZnO QDs with a quantum yields (QY) of 30% in ethanol were obtained. Benefiting from the amphibious property of HPEI, the ZnO/HPEI nanocomposites in ethanol could be dissolved in chloroform and water, acquiring a QY of 53%, chloroform and 11% in water. By this strategy, the ZnO/HPEI nano-composites could be applied in not only in optoelectronics, but also biomedical fields (such as bio-imaging and gene transfection). The bio-imaging application of water-soluble ZnO/HPEI nanocomposites was investigated and it was found that they could easily be endocytosed by the COS-7 cells, without transfection reagent, and they exhibited excellent biological imaging behavior.

## 1. Introduction

Semiconductor quantum dots (QDs) show unique size- and shape-dependent optical and electronic properties [1,2,3,4,5,6,7]. They are of great interest in various applications, such as optoelectronics [8,9], biophotonics [10,11,12,13,14,15], and nanomedicine [14,15,16]. Among these QDs, cadmium series QDs (such as CdSe [17], CdSe/CdS [18]) and zinc series QDs (such as ZnSe [19], ZnO [20,21,22]) often have excellent fluorescence. However, cadmium series QDs are actually toxic under certain conditions and free Cd^2+^ ions can be liberated from QDs, resulting in the cytotoxicity of QDs [23]. While, ZnO QDs are a cheap nanomaterial with non-toxic properties, they have great application in bio-medical fields, such as bio-imaging and drug release.

ZnO QDs are often prepared by the facile sol–gel approach [20,24,25,26] Zinc salts (such as Zn(OAc)_2_, Zn(NO_3_)_2_, zinc methacrylate), and alkali (such as LiOH, diethanolamine) are used as precursors. By varying the [Zn^2+^]/[OH^−^] molar ratio, the size tunability of defect rich ZnO QDs was achieved, resulting in size tunable visible emission of ZnO QDs, with the quantum yield as high as 76% [20,24]. Ionic liquid [27] and polymers [28,29,30], such as poly (ethylene glycol) methyl ether (PEGME) have also been used to prepare size-controlled ZnO QDs with quantum yield as high as 45%. Xiong et al. developed a two-step polymerization method, where they combined the polymerization or copolymerization of monomers (such as poly (ethylene glycol) methyl ether methacrylate (PEGMEMA) [31], methacrylic acid (MAA) [32], methyl methacrylate (MMA) [32], and methacrylamide [33]) with the in situ preparation of ZnO QDs to gain core–shell-structured ZnO@polymer QDs.

Until now, the polymers or monomers, used for ZnO quantum dot synthesis, always have carboxyl groups, ester groups, or amide groups. Therefore, the resulting ZnO/polymer nanocomposites cannot be applied in gene transfection because polymers with such groups cannot condense DNA well or at all. Hyperbranched poly(ethylenimine)s (HPEI), with a three-dimensional structure and abundant amine groups, have been regarded as one of the most promising non-viral gene vectors due to the proton sponge effect [34,35,36,37]. If HPEI are used as nanoreactors to synthesize fluorescent ZnO QDs, the advantages of HPEI (such as amphibious property, gene transfection and drug delivery capability) and ZnO QDs (such as fluorescence) could be integrated together, endowing the ZnO/HPEI nanocomposites with fluorescence, gene transfection, and amphibious properties. Herein, HPEI were applied as nanoreactors to synthesize amphibious ZnO QDs with blue fluorescence. The effect of [Zn^2+^]/[OH^−^] molar ratio and heating time on the fluorescence performance of prepared ZnO QDs was investigated. The amphibious property and fluorescence of the resulting ZnO/HPEI nanocomposites in different solvents were also followed. For the water-soluble ZnO/HPEI nanocomposites, they could easily be endocytosed by the COS-7 cells without other transfection reagent benefiting from the gene transfection property of HPEI and exhibited excellent biological imaging behavior.

## 2. Experimental Section

### 2.1. Materials

Hyperbranched polyethylenimines (HPEI) (Degree of Branching (DB) = 60%, *M*n = 1 × 10^4^, PDI = 2.5), zinc acetate (Zn(AC)_2_, 99.99%) and sodium hydroxide (NaOH, 99.99%) were bought from Sigma-Aldrich (Saint Louis, MO, USA). Ethanol and chloroform were purchased from Sinopharm Chemical Reagent Co. Ltd. (Shanghai, China). The ultrapure water with 18.2 MΩ·cm was used in all experiments.

### 2.2. Synthesis and Bio-Imaging Application

#### 2.2.1. Synthesis of HPEI/Zn^2+^

A totally of 384.0 mg of HPEI was dissolved by 30 mL of ethanol in a 250 mL conical flask. Then, the solution 13.0 mg of Zn(**O**AC)_2_ in 10 mL of ethanol was added. The mixture was vigorously stirred for 2 days.

#### 2.2.2. Synthesis of Amphibious ZnO QDs with Blue Fluorescence Based on HPEI Hyperbranched Polymers

Typically, 10 mL of HPEI/Zn^2+^ was deoxygenated and stirred in two-neck flask. Reflux was carried out for 2 h with oil bath heating at 100 °C under ordinary pressure. The heating temperature was then cooled down to 80 °C and 1.4 mg of NaOH ([Zn^2+^]/[OH^−^] = 1:2) in 4 mL ethanol and 1 mL ultrapure water was quickly injected into the reaction flask. Test samples were taken out from the reaction flask at 2, 4, 6, 8, and 10 h, respectively. The sample collected at 10 h was abbreviated as ZnO_1:2_/HPEI-10 nanocomposites. A similar procedure was also applied for other ZnO quantum dot synthesis under different [Zn^2+^]/[OH^−^] molar ratios such as 1:1 and 1:4.

#### 2.2.3. Bio-Imaging Application of ZnO/HPEI Nanocomposites with Blue Fluorescence

The original ethanol solution of ZnO/HPEI nanocomposites were firstly concentrated and then dialyzed by dialysis bag with molecular weight cut off (MWCO) of 1 k Da against distilled water to obtain an aqueous solution of ZnO/HPEI nanocomposites. The original culture medium of COS-7 cells was replaced by phosphate buffer solution (PBS) solution of ZnO/HPEI nanocomposites. After culturing for 6 h, the cell imaging of ZnO/HPEI nanocomposites were characterized by fluorescence microscopy (Olympus Optical Co., Tokyo, Japan).

### 2.3. Measurements

Transmission electron microscopy (TEM) and elemental characterization were performed on a FEI Tecnai F20 microscope (Thermo Fisher Scientific, Waltham, MA, USA), with an energy-dispersive X-ray spectrometer (EDS) (Thermo Fisher Scientific, Waltham, MA, USA), at an accelerating voltage of 200 kV. The ethanol solution of ZnO/HPEI nanocomposites was filtered by 220 nm polyethersulfone filter and sonicated for 10 s immediately before TEM sample preparation. TEM samples were made by adding 10 μL ethanol solution of ZnO/HPEI nanocomposites to a copper grid coated with ultrathin carbon film. UV-Vis spectra were characterized on a UV-1780 spectrophotometer with a 1 cm quartz spectrophotometer cell. For the quantum yield (QY) measurement, the solutions of ZnO/HPEI nanocomposites were commonly diluted with the corresponding solvents to make the absorbance at an excitation wavelength (360 nm) in the range of 0.03 to 0.1. Photoluminescence (PL) spectra were investigated by using a Cary Eclipse spectrometer (Agilent, Santa Clara, CA, USA) with an excitation wavelength of 360 nm. Dynamic light scattering (DLS) data were collected on a Zetasizer Nano-ZS (Malvern Instruments, Malvern, United Kingdom). Photographs were taken with a SAMSUNG WB150F digital camera (Samsung, Seoul, South Korea). TGA weight loss curves were acquired by using Netzsch STA-449F3 (Netzsch, Selb, Germany) under nitrogen at a heating rate of 20 °C/min from 30 to 800 °C.

## 3. Results and Discussion

HPEI have a large number of primary, secondary, and tertiary amines. Zn^2+^ ions were firstly complexed with the amine groups of HPEI, and then reacted with alkali, which was supplied by HPEI or addition. After refluxing for several hours, fluorescent ZnO QDs with blue emission were obtained based on HPEI, as shown in Scheme 1a–c. After dialysis, the ZnO/HPEI nanocomposites with blue emission were applied on bio-imaging, as shown in Scheme 1d.

The UV-Vis and photoluminescence spectra of ZnO QDs synthesized under different [Zn^2+^]/[OH^−^] molar ratios and various heating times are shown in Figure 1. For the ZnO QDs prepared under a [Zn^2+^]/[OH^−^] molar ratio of 1:2, the absorption peaks locate at approximately 325 and 383 nm. The absorption peaks have no significant change under various heating time, as shown in Figure 1a. The corresponding emission peaks of ZnO QDs, shown in Figure 1b, locate at 408–411 nm under different heating time. The inset photo shows the ZnO QDs emit blue light under UV light.

For the ZnO QDs prepared under [Zn^2+^]/[OH^−^] molar ratio of 1:1 and 1:4, the UV-Vis and PL spectra of ZnO QDs have no remarkable variation compared with those of ZnO QDs made under a [Zn^2+^]/[OH^−^] molar ratio of 1:2. This indicates that the heating time and molar ratios of [Zn^2+^]/[OH^−^] have no significant effect on the optical properties of ZnO QDs. For the common sol–gel approach used for ZnO QD synthesis [20,24,25,26], zinc salts and alkali were used to prepare size-controlled ZnO QDs without other stabilizers, and the hydroxyl ions and their counter ions actually as a capping layer around the QDs, thus, the size-tunability of ZnO QDs could be realized by [Zn^2+^]/[OH^−^] molar ratios. Here, HPEI, with a three-dimensional architecture and plenty of amines, were used to segregate, stabilize and control the size of ZnO QDs. The size-tunability of ZnO QDs might be tuned by adjusting the molar ratio of [Zn^2+^]/[HPEI], as reported in our previous work for Fe_3_O_4_ nanocrystal synthesis [37]. The hydroxyl ions and their counter ions cannot control the size ZnO QDs anymore, but they could still adsorb on the surface of ZnO QDs and influence the emission property of ZnO QDs.

The quantum yields (QY) of ZnO QDs were measured according to the method described in reference [38]. Cumarin 1 was chosen as a reference standard (QY = 0.73). The QY of ZnO QDs synthesized under different [Zn^2+^]/[OH^−^] molar ratios were monitored during the heating process and the results show that the QY of ZnO QDs, prepared under different [Zn^2+^]/[OH^−^] molar ratios, could all reach 30%, as shown in Figure 2. The QY of ZnO QDs prepared under a [Zn^2+^]/[OH^−^] molar ratio of 1:2 and various heating times showed better results compared with other conditions, indicating the hydroxyl ions and their counter ions virtually influence the optical properties of ZnO QDs, even in the presence of HPEI.

Transmission electronic microscopy (TEM) was used to characterize the morphology and size distribution of ZnO_1:2_/HPEI-10 nanocomposites, which were synthesized with [Zn^2+^]/[OH^−^] molar ratio of 1:2 and 10 h heating, as shown in Figure 3. It can be seen the ZnO QDs are relatively uniform and monodisperse. This is because HPEI could segregate ZnO QDs and prevent their aggregation. The ZnO QDs have a mean diameter of 3.0 nm by measuring hundreds of nanoparticles. The corresponding EDS spectrum, shown in Figure 3b, displays the signals of Zn and O elements, confirming the existence of ZnO QDs.

Figure 4 shows the size distribution of neat HPEI and ZnO_1:2_/HPEI-10 nanocomposites, measured by DLS. The hydrodynamic diameter of neat HPEI is 3.6 nm, while the ZnO_1:2_/HPEI-10 nanocomposites have a diameter of 4.9 nm. According to TEM data and the hydrodynamic diameter change of HPEI before, and after, quantum dot (QD) synthesis, the ZnO QDs should be encapsulated into the interior cavities of HPEI and one ZnO QD is complexed with each HPEI in average.

Figure 5 shows a comparison of FT-IR spectra between 4000 and 500 cm^−1^ of neat HPEI and ZnO_1:2_/HPEI-10 nanocomposites. The N–H stretching vibration of primary and secondary amine groups in HPEI locate at 3286 cm^−1^ in Figure 5a, and it red shift to 3429 cm^−1^ in Figure 5b for the ZnO_1:2_/HPEI-10 nanocomposites. The bands at 2917 (or 2921) and 2813 (or 2850) cm^−1^ in all curves correspond to asymmetric –CH_2_– stretching vibration, and symmetric –CH_2_– stretching vibration, respectively. The bending vibrations of primary amines and secondary amines, in HPEI in Figure 5a, appear at 1669, and 1589 cm^−1^, respectively. While, the bending vibrations of primary amines and secondary amines in ZnO_1:2_/HPEI-10 nanocomposites shift to 1631, and 1604 cm^−1^, respectively. These frequency changes in FT-IR can be attributed to the complexation interactions between amines of HPEI and ZnO QDs.

The composition of ZnO_1:2_/HPEI-10 nanocomposites was measured by thermogravimetric analysis (TGA), as shown in Figure 6. The decomposition temperature range of neat HPEI was 295–420 °C, which can be seen in Figure 6a. At 420 °C, the weight loss of HPEI is 100 wt%, indicating that HPEI have completely decomposed and volatilized. For the TGA of ZnO_1:2_/HPEI-10 nanocomposites, shown in Figure 6b, there is a certain amount of weightlessness of absorbed physics and chemistry water in the temperature range of 100–200 °C. Figure 6b also shows a large weight loss from 326 to 420 °C, which can be assigned to the decomposition of HPEI. After 420 °C, the weight loss is unchanged and the remained mass content, assigned to ZnO QDs, is about 3.2%. The content of ZnO QDs could be adjusted by changing the amount of Zn(OAc)_2_ and NaOH during ZnO QD synthesis. We have increased the amount of Zn(OAc)_2_ and NaOH to 196.6 mg (1.075 mmol), and 85.7 mg (2.150 mmol), respectively. While, the amount of HPEI (384 mg, 0.0154 mmol) is unchanged. The content of ZnO QDs could be increased to more than 30% under such synthesis condition.

Benefiting from the excellent solubility of HPEI in many solvents, such as ethanol, chloroform, water, dimethylsulfoxide (DMSO), N-methylformamide (NMF), the ZnO_1:2_/HPEI-10 nanocomposites should also be soluble in these solvents. Here, the original ZnO_1:2_/HPEI-10 ethanol solution was dried under vacuum, and then dissolved in ethanol and chloroform with a concentration of 3 mg/mL, respectively. The original ZnO_1:2_/HPEI-10 ethanol solution was also dialyzed by dialysis bag with molecular weight cut off (MWCO) of 1 k Da against distilled water to get water-soluble ZnO_1:2_/HPEI-10 nanocomposites. Figure 7 shows a comparison of UV-Vis and PL spectra of ZnO_1:2_/HPEI-10 nanocomposites in ethanol, chloroform, and water, respectively. The aqueous ZnO_1:2_/HPEI-10 nanocomposites have an obvious absorption peak at 365 nm, which is distinct from ZnO_1:2_/HPEI-10 nanocomposites in ethanol and chloroform phase, due to dialysis and solvent polarity differences. The corresponding emission peak has a red shift from 408 to 423 nm, compared with that of ZnO_1:2_/HPEI-10 nanocomposites in an ethanol and chloroform phase. For the fluorescence property, the QY of ZnO_1:2_/HPEI-10 nanocomposites in ethanol, chloroform, and water is 30%, 53%, and 11%, respectively.

HPEI, with a hyperbranched structure and many terminal amines, have been regarded as one of the most promising non-viral gene vectors. They are also widely used in uses, including gene transfection and drug delivery. While, ZnO QDs have excellent fluorescence and are non-toxic compared with cadmium-series QDs, the fluorescent ZnO/HPEI nanocomposites combine the advantages of HPEI and ZnO QDs together, and have great prospects on bio-imaging, gene transfection, and drug release with ZnO QDs as fluorescence probes. Here, the application of ZnO_1:2_/HPEI-10 nanocomposites on bio-imaging was investigated. The COS-7 cells showed no fluorescence under a fluorescence microscope. However, after the COS-7 cells were cultured for 6 h in the PBS solution of ZnO_1:2_/HPEI-10 nanocomposites, it can be clearly seen that blue fluorescence was observed under a fluorescence microscope, as shown in Figure 8. This phenomenon indicates that the ZnO_1:2_/HPEI-10 nanocomposites could be internalized by COS-7 cells via endocytosis. Green et al. reported that bare cysteine-capped particles could not be endocytosed by the cells and cationic liposome treated QDs can be internalized into human breast cancer cells [39]. It can be seen that the transfection reagent is essential for cell imaging applications. Benefiting from the gene transfection property of HPEI, the ZnO_1:2_/HPEI-10 nanocomposites could be easily endocytosed by cells.

## 4. Conclusions

In summary, water-soluble and oil-soluble HPEI were used as nanoreactors to synthesize uniform and amphibious ZnO QDs with blue fluorescence in ethanol. By changing [Zn^2+^]/[OH^−^] molar ratio and heating time, ZnO QDs with a QY of 30% were gained. The resulting ZnO/HPEI nanocomposites in ethanol could be dissolved in chloroform and water after vacuum drying, or dialysis, respectively. The QY of ZnO/HPEI nanocomposites can reach to 53% in chloroform and 11% in water. Thus, the application of the resulting ZnO/HPEI nanocomposites could be extended, not only to optoelectronics, but also biomedical field (such as bio-imaging, gene transfection). The resulting water-soluble ZnO/HPEI nanocomposites could easily be endocytosed by the COS-7 cells without transfection reagent and exhibited excellent biological imaging behavior.

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
