# Peer review of "In Situ Preparation of Amphibious ZnO Quantum Dots with Blue Fluorescence Based on Hyperbranched Polymers and their Application in Bio-Imaging"

_polymers, 2020, doi:10.3390/polym12010144_

Round 1
Reviewer 1 Report
The paper reports on the preparation of ZnO quantum dots using hyperbranched Poly (ethyleneimine). The hybrid material shows high solubility in organic and aqueous solvents. The authors suggest that such materials can be used in optoelectronic and in biomedical applications. However, the amount of the ZnO QDs is only 3% in the PEI matrix. Did the authors attempt to increase the QD content? Is this amount enough for the proposed applications? The authors should elaborate on these questions and include the relative discussion on their work. Additionally the references should be updated since there are recent publications in this matter that are missing.
Reviewer 2 Report
The paper titled "In situ preparation of amphibious ZnO quantum dots with blue fluorescence based on hyperbranched polymers and their application in bio-imaging" deals with the preparation on Quantum Dots based on Zinc Oxide and polyethyleneimines.
The paper is interesting and well written, I have only some minor points to be clarified
1) 2.1 Materials: HPEI is defined via DB and Mn but "DB" is not defined (I guess it is Degree of Branching) and Mn units are not given (should be Daltons). The same occurs in 2.2.3, where che MWCO is 1k with no units.
2) In 2.1 again, zinc acetate is indicated as Zn(AC)2 but usually zinc acetate is indicated as Zn(OAc)2.
3) 2.3 Measurements: measurements are not well specified. For example: how were UV-vis solutions prepared? concentrations, solvents etc.
How were TGA performed? Temperatures, gas, heating rate
How TEM samples were prepared?
This point must be largely improved
Round 2
Reviewer 1 Report
The authors have addressed all the comments so as the paper can be accepted in its present form.